# The Predictive Effect of Teachers' Perception of School Principals' Motivating Language on Teachers' Self-Efficacy via a Cultural Context

**Emir Ozeren** [1,*], **Aykut Arslan** [2], **Serdar Yener** [3]  **and Andrea Appolloni** [4,5,6,*]

[1] School of Tourism, Dokuz Eylul University, Izmir, Turkey & CISEI [Centre for Inclusive and Sustainable Entrepreneurship and Innovation], University of Southampton, Southampton SO17 1BJ, UK

[2] Department of International Business, Piri Reis University, 34940 Istanbul, Turkey; aarslan@pirireis.edu.tr

[3] Department of Business Management, Boyabat School of Business, 57200 Sinop, Turkey; serdaryener@sinop.edu.tr

[4] Department of Management & Law, University of Rome 'Tor Vergata', 00133 Rome, Italy

[5] School of Management, Cranfield University, Cranfield, Bedford MK43 0AL, UK

[6] Institute for Research on Innovation and Services for Development, National Research Council, 80134 Naples, Italy

[*] Correspondence: emir.ozeren@deu.edu.tr (E.O.); andrea.appolloni@uniroma2.it (A.A.)

**Abstract:** The aim of the study is to investigate the relationship between teachers' perception of school principals' motivating language and teachers' self-efficacy mediated by the cultural context. School principals' linguistic communication skills are critical to sustain the motivation of teachers and their self-efficacy. Motivating language theory (MLT), on which this study is based, provides a model that helps us understand how the language, more precisely, the speech acts, used by school principals have an impact on teachers' self-efficacy. A survey method was employed with 252 teachers through convenience sampling. The teachers' mean age was 34.87 (SD = 9.22) years, and the average length of service was 11.72 (SD = 9.42) years. The results showed that school principals' use of motivating language was significantly and positively associated with teachers' self-efficacy (b = 0.10, $p < 0.000$). The low-context culture was found to have a full mediating effect in this relationship (b = 0.04, t = 3.1771, $p < 0.000$). The findings contribute to leader communication theory by highlighting a particular emphasis on the language school principals use to motivate teachers.

**Keywords:** motivating language theory; self-efficacy; teacher motivation; school principal; administrator; verbal leadership; communication; education for sustainable development

## 1. Introduction

In today's world, communication is one of the most sought-after skills for those in managerial and leadership positions [1]. Previous studies have shown that people in leadership roles spend about 80% of their day communicating in some form or another [2]. In a study on Swedish CEOs, Tengblad [3] found that the CEOs spent 62% to 80% of their time performing verbal leadership work. Within the area of educational leadership, school principals spend up to 84% of their time engaged in verbal leadership communication [4]. These rates will continue to increase due to current developments in communication technologies and the recent surge of interest in how leaders use communication as a tool to motivate their followers [5,6]. It is highly likely that ineffective communication leads to low performance, whereas inspirational communication produces sustainable results, which exceed expectations [7]. Leaders' strategic use of verbal communication in motivating forms enhances followers' satisfaction and sustainable performance [8] and significantly improves employee behaviours [1,9] and subordinates'

outcomes [10]. A recent study by Binyamin and Brender-Ilan [11] provides strong evidence that effective leaders use their communication skills and adopt motivating language as a sustainable mechanism to motivate employees with the goal of achieving organisational outcomes. However, the particular emphasis on the language used by leaders to motivate their followers has received relatively less attention than either communication or leadership scholarship [12]. To date, few attempts have been made to examine the impact of language use on leaders' effectiveness in communicating with their followers in general [13] and the specific language used by school principals in motivating teachers in schools in particular.

The motivating language theory (MLT), which is the main theory on which this study is based, describes the effect of the language and, more precisely, the speech acts, strategically and intentionally used by leaders to enable organisations to reach expected sustainable outcomes [9,13]. Given that leaders spend most of their time communicating with their followers, the strategic use of speech acts [8] has become critical to obtain results within an organisation. A preliminary glance at the vast literature on leadership studies reveals that the research is largely focused on leaders' behaviour, characteristics, skills, and leadership styles without an adequate understanding of the everyday language they use [9,12,14]. This insufficiency is paramount in school settings where school principals' linguistic communication skills are critical to the motivation of teachers and their self-efficacy in the classroom [1,9]. A recent study by Moradkhani and Haghi [15] indicated that verbal persuasion is the most powerful source in enhancing teachers' self-efficacy in the Iranian context.

As far as the effect of leaders' use of ML on teachers' self-efficacy, a gap exists in the educational research literature; as a potentially important link, this relationship deserves further scholarly attention in the context of how school principals can motivate teachers. To the best of our knowledge, no research has thus far explicitly measured the effect of school principals' use of ML on teachers' self-efficacy via cultural context. The present study is an attempt to partially address this gap. In this article, the culture was framed by [16] classification of high and low context based on teacher participants' own styles of communication and their shared understanding and perception regarding the ML use by their principals. The current study aims to contribute to the relevant stream of literature by investigating whether teachers are motivated by the daily language school principals use in communicating with them and examining the mediating role of cultural context in the relationship between ML and teachers' self-efficacy. Based on the results of this study, school principals are expected to support the development of teachers using motivating language within the appropriate cultural context in order to contribute to education for sustainable development. Despite the fact that a significant body of literature and discussion surrounding education for sustainable development exists, the language and linguistic communication skills school principals have to sustain the motivation of teachers and enhance their self-efficacy has received scant attention. In our study, we partially address this gap in the literature by clearly highlighting the aspects of motivational language use by school principals to enhance teachers' self-efficacy in the light of cultural context. School principals' influence on teachers through motivational language use is substantial as the level of teachers' self-efficacy increases, teachers' burn out and attrition can be lessened and hence, this may result in more motivated, committed, and sustained workforce of teachers as well as higher students' academic achievement [17,18] and teaching effectiveness [19] that are all likely to contribute to education for sustainable development. While education for sustainable development enables people to understand the influences of their own actions by encouraging them to take responsible decision [20], enhancing teachers' self-efficacy through motivational language use by school principals can serve this purpose as teachers with high self-efficacy hold the sense of being able to achieve a certain task in a specific situation. More specifically, teachers with high self-efficacy believe that they can teach even the most difficult and unmotivated students so that they can improve student achievement [21].

The paper is organised as follows: The second section presents the literature review in six subheadings that are 'motivating language theory', 'self-efficacy in teaching', 'relationship between motivating language and self-efficacy', 'cultural context', 'relationship between motivating language and

cultural context', and 'the mediating role of cultural context'. The third section explains the methodology part in greater details including research design, sample, and data analysis ('common method variance', 'measures', 'validation of the measures', 'results of hypothesis tests'). The fourth section discusses the results in the light of literature. The fifth section presents practical implications derived from this study. The sixth section offers suggestions for future research. The last section presents the concluding remarks of the manuscript.

## 2. Literature Review

### 2.1. Motivating Language Theory

Motivating language theory (MLT), on which this study is based, provides a model that allows us to explain how the language of leadership works and how it affects a leader's followers. First conceptualised by Sullivan [8], MLT was later reframed by other researchers [9]. In this study, we used the psycholinguistic analytical framework proposed by Sullivan [8] to examine three basic acts of speech used by leaders towards their followers. The first is the perlocutionary act, a business-oriented method of communication that reduces uncertainty and gives direction and orders (i.e., it defines the job to be done). The second is the locutionary act that occurs when a leader's language to his or her members explains the structure, rules, and values of an organisation's culture by making meaning and using stories. This means of communication is used to grant a symbolic status to the working environment; in this way, employees make sense of their workplace, and their behaviours can meet organisational objectives. The third act is the illocutionary act, or empathetic language, which is a method of communication that improves relationships and interpersonal ties. Different from the perlocutionary act, the illocutionary act does not involve giving direction on how to perform a task, but rather exists as an informal behaviour at the human level between managers and subordinates or leaders and members.

Sullivan [8] indicated that impressive managers use all three types of ML to achieve their goals, noting the relationship between desired outcomes and the specific forms of ML that need to be used [22]. Similarly, Sarros et al. [13] reported in their exploratory qualitative study that leaders develop a repertoire, which includes linguistic approaches to achieve organisational outcomes. MLT explains the ways in which managers who are successful at motivating workers. Such managers not only provide the knowledge and feedback their workers need, but also help them stand behind the organisation's values and objectives by developing a habit of improving informal communication with them. Supportive acts of speech when managers try to establish real lines of communication with their employees play a role in increasing workers' motivation [8] (p.113).

### 2.2. Self-Efficacy in Teaching

Within the framework of social cognition theory, the concept of self-efficacy is defined by Bandura [23] as an individual's belief upon starting an act that they will be effective in that specific situation and thus maintain the act until achieving a result. Self-efficacy can be defined as "a perceived capability to perform target behaviour" [24]. It is important to emphasise that self-efficacy does not correspond to a person's actual skills but rather to the amount of faith, the person has in their skills. People with high self-efficacy accept a new situation as a challenge and do not give up easily when failing, whereas people with low self-efficacy do exactly the opposite and tend to quickly abandon difficult tasks. There are four major sources that contribute to the development of self-efficacy [23]: Personal accomplishments; vicarious experience; verbal persuasion, or advice and encouragement for an individual to achieve something; and psychological states.

In an educational setting, teachers' self-efficacy can be conceptualised as 'teachers' beliefs in their own abilities to plan, organise, and carry out activities required to attain given educational goals' [25] (p. 153). Expectancy of sustainable outcomes and teachers' self-efficacy beliefs interact to predict motivation of teaching behaviour [26]. Several studies have successfully provided evidence of a positive

relationship between school principals' instructional leadership styles and teachers' perceptions of self-efficacy [27–29]. A recent study examined 82 empirical studies and specifically looked at how researchers measured and conceptualised sources of teaching self-efficacy [30]. The findings revealed more problems than answers, due to a few methodological shortcomings.

Traditional teaching methods are gradually being replaced by interactive, supportive, and motivating methods through which students internalise what they are learning [31]. Today's educators are responsible for teaching students how to learn and serving as a model throughout this process. In learning environments where leaders' expectations and their supportive and motivating behaviours are communicated clearly, the questioning role of teachers in the process whereby students attempt to obtain knowledge may help students achieve this behaviour. Studies show that teacher self-efficacy can be related to and predict teachers' different cognitive, emotional, and behavioural responses; for example, it is correlated positively with teacher job satisfaction and job commitment, but negatively with burnout and teacher attrition [25] (p. 101). High teacher self-efficacy is considered to potentially lead to positive student outcomes [32,33]. Teachers with high self-efficacy can plan effective instructional strategies, increase sustainable performance, and enhance their effectiveness and productivity [34] (p. 79). These teachers can also help students with academic adjustments as well as contribute to sustainable classroom quality [35]. Moreover, teachers with high self-efficacy are important in the adaptation and upgrading of the education process according to the changing needs of the era [36]. There is a large body of evidence supported by multivariate meta-analyses on teachers' self-efficacy. Accordingly, classroom management self-efficacy of teachers was significantly and negatively associated with burnout [21,37] and positively associated with students' academic achievement [17]. Teachers' self-efficacy was not only found to enhance students' academic achievement but also overall teaching effectiveness [19]. In predicting commitment to the teaching profession of preservice and inservice teachers, their self-efficacy beliefs can be utilized [38].

### 2.3. Relationship between Motivating Language and Self-Efficacy

Bandura [23] conceptualized self-efficacy beliefs through a four-dimensional model (mastery experiences, verbal persuasion, vicarious experiences, and physiological arousal). From the teacher's perspectives, the verbal persuasion "has to do with verbal interactions that a teacher receives about his or her performance and prospects for success from important others in the teaching context, such as administrators, colleagues, parents, and members of the community at large" [39] (p. 945). The research on self-efficacy [1,2,40] has pointed to the particular importance of language and verbal persuasion among the four sources of self-efficacy, stating that ML might play a role in developing self-efficacy, inasmuch as people tend to exert more effort when they are encouraged in real terms [41] (p. 365). Another study found that verbal persuasion increased the self-efficacy of pre-service teachers by means of representative experience [42].

Motivating language has been 'shown to be a consistent predictor of major workplace outcomes' [43]. For example, Mayfield and Mayfield [Mayfield, J.; Mayfield-2010] examined the role of ML in the relationship between self-efficacy and performance and identified that employee self-efficacy played a partial mediating role in the relationship between leaders' motivating language and employee performance. In other studies, ML was shown to influence employees' perceptions of the creative environment [43] and to enhance employees' decision-making [9]. In another recent study, ML was found to be associated with promoting positive public-school climates in Kuwait [44]. The potential of verbal persuasion to increase self-efficacy is related to the Pygmalion effect, namely, that adjusting one's behaviour appropriately to a situation contributes to achieving the desired results. In their study, which demonstrated the observer-expectancy effect, [45] told teachers that a certain group of students had high IQ levels, while another group included students with low IQ levels. Teachers were observed to spend more time with the students with high IQ levels and developed tasks that are more challenging for them. The success of the Pygmalion effect depends on both the leaders' ML use and verbal persuasion and the credit given by leaders to the workers they expect to be successful.

The following studies in the Turkish literature were also reviewed carefully to give greater insight into the context in which the current study was conducted: Karaaslan [46] examined the mediating effect of leader–member interaction on the relationship between ML and organisational citizenship behaviour. Mert et al. [47] examined the effect of managers' ML on employees' perceptions of their own and their managers' performance. Özen [48] investigated the interaction of the use of ML by school principals and the organisational citizenship behaviours of teachers. Finally, Karademir and Ergeneli [49] examined the effect of perceptions of psychological empowerment on the interpersonal trust between managers and employees and on perceived employee performance and assessed the role of ML on this relationship. Based on the objective of this study that is to examine the effect of teachers' perception of school principals' motivating language on teachers' self-efficacy via their cultural contexts along with aforementioned robust evidence in the literature, the first hypothesis of this study is as follows:

**Hypothesis 1 (H1).** *Principals' ML is positively associated with teachers' perceptions of self-efficacy.*

### 2.4. Cultural Context

Another important factor worth considering in the relationship between the use of ML by school principals and teachers' self-efficacy is cultural context [50–53]. Actually, the role of the context of a speech act considered as constructed has taken its roots in the earlier and pioneering studies in the philosophy of language [54–56].

Communication is a way for individuals to express themselves to others and refers to the making sense of a message sent by a sender to a recipient. The sender may encode a message using words, gestures, mimics, and behaviours. The ability of a recipient to decode the message and receive the meaning intended by the sender is directly proportional to the sender's and the recipient's shared background. In this decoding process, dimensions of a common culture effectively allow for the correct sense of a message to be understood. Messages can be interpreted differently by individuals from each culture around the world [57]. Culture shapes our perceptions and disciplines our conscious and subconscious [16] (p. 87). Knowledge of the characteristics of a culture can enable an observer to guess the behaviours of individuals from these cultures. Soley and Pandya [58] defined culture as a shared system of perceptions about history, religion, traditions, customs, beliefs, behaviours, and so forth.

Hall [16] divided cultures into low-context and high-context depending on their methods of communication and other characteristics. High-context culture refers to the culture of societies with a strict hierarchy, strict rules, and behavioural norms in which the meaning in communication cannot be solely expressed verbally or in writing, and communication is interpreted based on the parties' previous experiences and perceptions. In a low-context culture, meaning is expressed by direct and clear communication [59]. Low-context cultural traits are found in individualistic societies, while high-context cultural traits are seen in collectivist societies. Individualistic societies are expected to have high self-efficacy due to their clear and direct communication behaviours, whereas collectivist societies are expected to have low self-efficacy because of their indirect and implicit communication. Ultimately, communication depends on individuals' perception, interpretation, and assessment of verbal and nonverbal behaviours [57].

As earlier studies have reported [1,3,60], Turkey exhibits high-context cultural characteristics, which indicates that what school principals intend to say is just as important as what they actually say for establishing effective communication. This duality reveals the need to include other environmental factors and cues (style, gestures, mimics, body language, and stress, etc.) in the context of the language used by school principals in communicating with teachers. If individuals prefer implicit and indirect communication (high-context culture), they may not be content with what has been said and instead must look for different contexts (previous experience, tone of voice, facial expressions, hints, etc.) to understand what was intended to be said [61] (p.18).

Table 1 summarises the communication preferences of societies with high- and low-context cultures.

**Table 1.** Communication ways of societies with high and low-context cultures adapted from Erdem [62] (p. 19).

| High-Context Culture | Low-Context Culture |
| --- | --- |
| <ul><li>Implicit, implicative, indirect</li><li>Environmental factors (physical environment, previous experiences, tone of voice, facial expressions, body language, etc.) are required.</li><li>What is said is different from what is intended to be said</li><li>Message has different meanings depending on people and environments (subjective communication)</li></ul> | <ul><li>Clear, explicit, direct</li><li>Little need for environmental factors (physical environment, previous experiences, tone of voice, facial expressions, body language, etc.)</li><li>What is said is in parallel with what is intended to be said</li><li>Message has the same meaning for all in all environments (objective communication)</li></ul> |

### 2.5. Relationship between Motivating Language and Cultural Context

Conger [50] divided the ML of leadership into two skill categories: The first is the process of meaningfully defining the organisational goals; it refers to the leader's message and is called 'framing'. The second skill is the leader's ability to use symbolic language to lend emotional power to their message to the members. This process is called 'rhetorical crafting'. The message provides a sense of direction to the leaders through framing, and the leader motivates the members with rhetorical symbols. Leaders provide the organisation with a mission and vision using framing; thus, they reduce uncertainty and remove confusion from employees' minds by increasing their adaptation to the process. In this way, the leaders pave the way for transparency and information sharing in the organisation process by clearly specifying the mission, vision, and goals. It is possible that framing can also help reduce uncertainty and misunderstandings, which are among the negative effects of a high-context culture or creating learning environments with low-context culture. Through framing and rhetoric crafting, the use of ML can provide learning environments, which include questioning, discussion, and positive criticism, the characteristic behaviours of a low-context culture [50]. We argue that the implicit and indirect communication of high-context cultures may not be experienced in a motivating process where expectations from the learning process and outcomes are expressed clearly and teachers are supported by verbal or nonverbal behaviours. Therefore, we can make the following hypotheses:

**Hypothesis 2 (H2a).** *Principals' use of ML is positively associated with teachers' low-context culture.*

**Hypothesis 2 (H2b).** *Principals' use of ML is negatively associated with teachers' high-context culture.*

### 2.6. The Mediating Role of Cultural Context

Markus and Kitayama [51] claimed that self-efficacy perceptions can be improved by developing personal traits and an autonomous sense of self, which are considered valuable qualities in individualistic societies, and thus that self-efficacy perception is specific to individualistic societies. It was indicated that increased self-efficacy perception in collectivist societies depends on effective intragroup communication [58]. However, Bandura et al. [63] suggested that self-efficacy beliefs solved the adaptation problem in both individualistic and collectivist societies by functioning as a regulative process.

Yin and Kuo [64] conducted a study on organisational communication problems and revealed how direct and indirect speech acts influence employee comprehension of process. They suggested that leaders use clear language to meet employees' informational and other needs that are based on interpersonal respect. In this way, employee comprehension and internalisation of the process can be ensured. In such organisations, direct and clear expression of a message can help employees understand the organisation's mission and vision and conform to its expectations while working. In addition, leaders' use of clear, respectful, and direct communication with employees provides an environment that favours the sharing of work-related information and experiences and the opportunity to increase employees' effectiveness and their interaction with each other [63]. In this sense, low-context culture

is considered to increase self-efficacy, since verbal persuasion or clear communication behaviour, which are among the components of self-efficacy theory, are more common in societies with low-context cultures than high-context ones. Moreover, indirect and implicative communication in low-context cultures is believed to reduce self-efficacy. Thus, we hypothesise the following:

**Hypothesis 3 (H3a).** *Teachers' low-context culture is positively associated with their self-efficacy perceptions.*

**Hypothesis 3 (H3b).** *Teachers' high-context culture is negatively associated with their self-efficacy perceptions.*

Redmond [65] indicated that encouragement, and conversely discouragement, plays an important role in increasing or reducing self-efficacy. Negative expressions from the people around them cause individuals to feel anxiety, stress, and disappointment. If leaders' use of ML is considered the only factor that increases self-efficacy of individuals, and the other contextual features of cultures are ignored, we cannot explain the fact that the use of ML by leaders produces different results in different societies [66]. A culture's contextual features considerably affect the impact of the language used by leaders on employees. In this sense, Hall and Hall [66] argued that the communication culture of a society provides a reference point for the level of interaction amongst society members. For instance, the time and place of communication and the nonverbal communication tools named as 'silent language' complement the indirect and implicit communication of high-context cultures [66]. In high-context cultures, the correct interpretation of messages is possible through these complementary factors. On the other hand, in low-context cultures, the correct interpretation of the messages conveyed by leaders depends solely on their correct encoding and decoding. As a result, communication features of low-context cultures play a critical role in developing employee self-efficacy beliefs.

Based on the abovementioned facts, we believe that the use of ML can increase teacher self-efficacy through clear and interactive communication within the school setting. The reason is that the opportunity for employees to ask for a leader's or other employees' help whenever they need it will increase their self-efficacy in low-context cultural environments that provide clear, direct, and interactive communication. Having access to such opportunity and using it on their own will increase the employees' self-confidence. The use of ML by leaders is beyond the control of employees and depends on the leaders' personal traits and is thus not a source that employees can always apply to. From this perspective, the ML of leaders is believed to be ineffective in the absence of a low-context culture that provides clear and direct communication. Based on these statements, we posit the following:

**Hypothesis 4 (H4a).** *Teachers' low-context culture acts as a mediator of the relationship between school principals' use of ML and the teachers' self-efficacy beliefs.*

**Hypothesis 4 (H4b).** *Teachers' high-context culture acts as a mediator of the relationship between school principals' use of ML and the teachers' self-efficacy beliefs.*

Based on the theoretical framework summarised above, the research model of the study is designed as where the independent variable is teachers' perception of school principals' use of ML and the dependent variable is teacher's self-efficacy. Low- and high-context cultures are used as mediating variables. Figure 1 shows the research model.

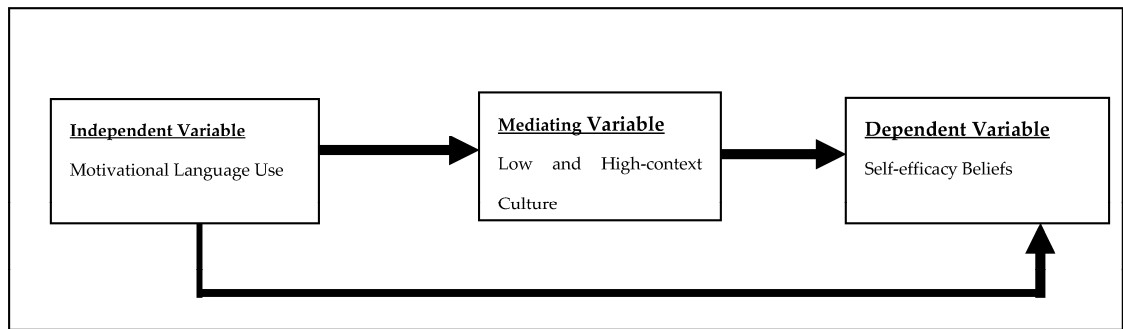

**Figure 1.** Research model.

## 3. Method

### 3.1. Design

A quantitative research approach, the survey method was adopted in the study. We contacted our online network (e.g., blogs) of school principals where teachers share opinions and discuss their matters. Data collection was performed between 20 July 2017, and 31 October 2018. By the end of the data collection process, 252 usable survey forms were collected. Elementary (grades 1–4), secondary (grades 5–8), and high school (grades 9–12) teachers were included in the study. The convenience sampling method was conducted. The data were analysed using Process 3.1 macro in SPSS and JAMOVİ 9.5.17, a freely available analysis tool downloadable from jamovi.org. To reveal the mediating relationships, a parallel mediating model (Model 4) was chosen.

### 3.2. Sample

Before testing the hypotheses, the descriptive statistics for the sample were estimated and are shown in Table 2 Of all the teachers who filled out the forms, 37.70% ($n = 95$) were male, and 62.30% were female ($n = 157$). The teachers' mean age was 34.87 (SD = 9.22) years, and the average length of service was 11.72 (SD = 9.42) years. The teachers' educational background showed that 61.11% ($n = 154$) had a bachelor's degree; 30.16% ($n = 76$) had a master's degree; 7.94% ($n = 20$) had a PhD degree; and 0.8% ($n = 2$) had an associate's degree. The study took place in public schools only. The number of teachers working at elementary level schools (grades 1–4) were 48 (19%); at secondary level schools (grades 5–8) were 84 (33%), and at high school level (grades 9–12) were 120 (48%) (see Table 2).

**Table 2.** Descriptive statistics.

| Variables | X | SD |
|---|---|---|
| Age | 34.87 | 9.22 |
| Tenure | 11.72 | 9.42 |
| **Gender** | **n** | **%** |
| Male | 95 | 37.70 |
| Female | 157 | 62.30 |
| **Education** | **n** | **%** |
| Assoc. Degree | 2 | 0.8 |
| Bachelor's | 154 | 61.11 |
| Master's | 76 | 30.16 |
| PhD | 20 | 7.94 |
| **School Types** | **n** | **%** |
| Elementary (1–4) | 48 | 19 |
| Secondary (5–8) | 84 | 33 |
| High school (9–12) | 120 | 48 |

*3.3. Data Analyses*

3.3.1. Common Method Variance

To reduce the possibility of common method bias (CMB) we follow three procedural remedies. First, we shared the survey form including only self-efficacy and motivational language scales. Following 4 weeks in the next round of data collection, we distributed surveys including cultural context variables [67]. This temporal, psychological, or methodological separation of measured items/measurement of variables is suitable to reduce CMB when the researcher is unable to collect data from multiple sources (p. 146). Second, we checked the VIFs and tolerance values. These were below 3 and the tolerance values were below 1, indicating the absence of CMB [68]. Lastly, we ran a Harman's Single-Factor Test. In this test all items (measuring latent variables) were loaded into one common factor. It is expected that the total variance for a single factor should be less than 50%; our test results indicated a total variance of 41, 95%. Thus, we conclude that CMB was minimized as much as possible both in the administration process and after post-hoc analyses.

3.3.2. Measures

All statements were rated on a five-point Likert scale ranging from 1 (strongly disagree) to 5 (strongly agree).

Motivating language scale: The scale was developed by Mayfield and Mayfield [69] and adapted into Turkish by Özen [48]. It comprises three factors and 24 questions. The following is a sample item: "Provides me with easily understandable instructions about my work".

General self-efficacy scale: The scale was developed by Sherer et al. [70] and adapted into Turkish by Yıldırım and İlhan [61]. The following is a sample item: 'I can usually handle whatever comes my way'.

Cultural communication scale: The scale was developed by Erdem [62] in accordance with Turkish culture largely based on Hall's model. The following is a sample item: 'While communicating in our society, one should pay attention to what is implied but not to what is said'.

The mean scores of latent variables were used in the study; thus, factors were irrelevant for the analyses. However, it is advised to report how many factors are extracted and see if the scales fit to their originals with this regard.

Validation of the Measures

To validate the proposed model, the psychometrics and the fit measures were calculated. Discriminant and convergent validity were tested, and the obtained values were compared with the acceptable values proposed in the literature. Table 3 shows the extracted factors and variance along with KMOs and reliability scores. For each latent variable, confirmatory factor analyses were also performed. The three-factor model of general self-efficacy scale yielded loadings ranging from 0.57 to 0.79 (the CFI is 0.92; TLI is 0.89; and the RMSEA is 0.07); for culture scale, we obtained two factors (low and high-contexts) and the factor loadings ranged from 0.50 to 0.78 (the CFI is 0.93; TLI is 0.94; and the RMSEA is 0.06); and lastly, like the original scale, we extracted three factors for Motivating Language scale and the factor loadings varied from 0.59 to 0.91 (the CFI is 0.92; TLI is 0.90; and the RMSEA is 0.06). All the CFAs appear to have an acceptable fit to the data.

To test the discriminant validity, first the average variances extracted were examined. Table 3 shows the descriptive statistics, standard deviations, and inter-construct correlations. The average variances explained are shown in bold and diagonal and the average variances explained are larger than the correlation coefficients of the latent variables.

**Table 3.** Psychometrics and fit measures.

| Variables | Ext Factors | Ext Variance [min-max] | KMO | Reliability | CFI | TLI | RMSEA | $\chi^2$ | df | $p$ |
|---|---|---|---|---|---|---|---|---|---|---|
| SE | 3 | 0.57–0.79 | 0.84 | 0.80 | 0.92 | 0.89 | 0.07 | 70.3 | 32 | <0.001 |
| LoCont HiCont | 2 | 0.50–0.78 | 0.72 | 0.83 0.87 | 0.93 | 0.94 | 0.06 | 956 | 45 | <0.001 |
| ML | 3 | 0.59–0.91 | 0.94 | 0.90 | 0.92 | 0.90 | 0.06 | 508 | 116 | <0.001 |

SE = Self-efficacy; LoCont = Low-Context and HiCont = High-Context; ML = Motivating Language.

Results reported in Table 4 showed that motivating language was positively correlated with both general self-efficacy (r = 0.21, $p$ < 0.01) and low-context culture (r = 0.25, $p$ < 0.01); but it was negatively correlated with high-context culture (r = −0.40, $p$ < 0.01). Motivating language was significantly associated only with one sub-factor of self-efficacy, initiating (r = 0.42, $p$ < 0.01), but not with the other sub-factors. Similarly, the sub-factors of motivating language (meaning making, emphatic, and direction giving language) were all positively correlated only with initiating (r = 0.41, r = 0.36, r = 0.37, $p$ < 0.01, respectively).

**Table 4.** Means, standard deviation, AVEs (Average Variance Extracted), and inter-construct correlations.

| Variables | X | SD | SE | In | Res | Pers | LoCont | HiCont | ML | MML | EL | DGL |
|---|---|---|---|---|---|---|---|---|---|---|---|---|
| SE | 4.24 | 0.47 | [0.61] | | | | | | | | | |
| In | 4.26 | 0.62 | 0.641 ** | NA | | | | | | | | |
| Res | 4.21 | 0.68 | 0.773 ** | 0.257 ** | NA | | | | | | | |
| Pers | 4.25 | 0.68 | 0.735 ** | 0.168 ** | 0.387 ** | NA | | | | | | |
| LoCont | 4.06 | 0.71 | 0.358 ** | 0.447 ** | 0.171 ** | 0.169 ** | [0.51] | | | | | |
| HiCont | 3.16 | 1.01 | −0.158 * | −0.059 | −0.071 | −0.206 ** | −0.077 | [0.58] | | | | |
| ML | 3.20 | 1.03 | 0.212 ** | 0.424 ** | 0.020 | 0.033 | 0.245 ** | −0.398 ** | [0.74] | | | |
| MML | 3.18 | 1.09 | 0.177 ** | 0.414 ** | −0.007 | −0.003 | 0.191 ** | −0.333 ** | 0.905 ** | NA | | |
| EL | 3.06 | 1.24 | 0.223 ** | 0.356 ** | 0.040 | 0.100 | 0.229 ** | −0.411 ** | 0.877 ** | 0.664 ** | NA | |
| DGL | 3.36 | 1.12 | 0.165 ** | 0.373 ** | 0.018 | −0.016 | 0.237 ** | −0.319 ** | 0.907 ** | 0.787 ** | 0.665 ** | NA |

SE = Self-efficacy; In = Initiate; Res = Resilience; Pers = Persistence; LoCont = Low-Context and HiCont = High-Context; ML = Motivating Language; MML = Meaning Making Language; EL = Emphatic Language; DGL = Direction Giving Language, *n* = 252, Correlation is significant at the 0.01 level (2-tailed). ** and at the 0.05 level (2-tailed). * AVEs in the diagonal.

Second, as presented in Table 5, we conducted the VIFs (variance inflation factors), and tolerance values to see if there is a multicollinearity problem. The VIFs were below 3 and the tolerance values were below 1. Finally, we checked the Cronbach's alpha reliability coefficients. They are the indicators of internal consistency and it was proposed to be at least 0.70 or higher [71]. As shown in Table 2, the values range between 0.80 and 0.90.

**Table 5.** Variance inflation factors and tolerance values.

| Variables | VIFs | Tolerance |
|---|---|---|
| LoCont | 1.065 | 0.939 |
| HiCont | 1.189 | 0.841 |
| ML | 1.258 | 0.795 |

LoCont = Low-Context and HiCont = High-Context; ML = Motivating Language.

### 3.3.3. Results of Hypothesis Tests

Table 6 shows the regression coefficients, standard errors, and model summary information of parallel multiple mediator model. From a parallel multiple mediation analysis conducted using ordinary least squares path analysis (using 5000 bootstrap samples), we tested if the motivating language (ML) of school principals indirectly influenced general self-efficacy of the teachers through their high and low context cultural values. Starting first with the total effect, as can be seen in Table 6, we found that principals' ML is positively associated with teachers' perceptions of self-efficacy

(c = c′ + (a1b1) + (a2b2) (c = 0.043 + 0.018 + 0.037 = 0.098)). Second, we tested if the principals' use of ML is positively associated with teachers' low-cultural context cultural and negatively associated with teachers' high-context cultural values (a2 and a1). The results showed that both assumptions are correct (a2 = 0.17, $p < 0.000$, and a1 = −0.39, $p < 0.000$). In addition, the recommended models explain 16% and 6% of the variance in teachers' self-efficacy beliefs, respectively. Third, we also investigated if the teachers' low-context cultural values are positively and if their high-context cultural values are negatively associated with their self-efficacy perceptions (b2 and b1). The tests yielded significant results only for teachers from low-context cultural background (b2 = 0.22, $p < 0.000$). As for teachers from high-context culture, the results were insignificant (b1 = −0.05, $p > 0.000$). This time the recommended model explains 15% of the variance in teachers' self-efficacy.

Lastly, we predicted if principals' use of ML was effective on teachers' self-efficacy beliefs indirectly through both low and high-context cultural values (mediating effects). The first mediating analysis, low-context cultural values', is estimated as a2b2 = 0.168 × 0.221 = 0.037 indicating a full mediating effect (b = 0.04, t = 3.1771, $p < 0.000$). Second analysis, high-context cultural values', is estimated as a1b1 = −0.389x − 0.045 = 0.018 indicating no mediating effect (b = 0.02, t = 1.4503, $p > 0.000$).

Thus, we found evidence that supported our assumptions for H1, H2a, H2b, H3a, H3b, and H4a but not for H4b.

**Table 6.** Regression coefficients, standard errors, and model summary information of parallel multiple mediator model.

| Antecedent | | $M_1$ [HiCont] | | | | $M_2$ [LoCont] | | | | Y [SE] | | |
|---|---|---|---|---|---|---|---|---|---|---|---|---|
| | | Coeff. | SE | p | | Coeff. | SE | p | | Coeff. | SE | p |
| X [ML] | $a_1$ | −0.389 | 0.057 | <0.000 | $a_2$ | 0.168 | 0.042 | <0.000 | $c'$ | 0.043 | 0.030 | 0.158 |
| $M_1$ [HiCont] | | | | | | | | | $b_1$ | −0.045 | 0.030 | 0.136 |
| $M_2$ [LoCont] | | | | | | | | | $b_2$ | 0.221 | 0.041 | <0.000 |
| Constant | $iM_1$ | 4.405 | 0.190 | <0.000 | $iM_2$ | 3.523 | 0.141 | <0.000 | $i_y$ | 3.349 | 0.213 | <0.000 |
| | | $R^2 = 0.159$ | | | | $R^2 = 0.060$ | | | | $R^2 = 0.152$ | | |
| | | F[1.250] = | 47.135 | $p < 0.000$ | | F[1.250] = | 16.026 | $p < 0.000$ | | F[3.248]= | 14.857 | $p < 0.000$ |

SE = Self-efficacy; LoCont = Low-Context and HiCont = High-Context; ML = Motivating Language.

## 4. Discussion

The increasing expectations placed on teachers can lead to requirements of further qualifications in their subject areas. Due to teachers' acceptance as the main actors of education, expectations of them are not only increasing but also diversifying. A school's performance may depend on its teachers' qualifications. Through effective interactions with school principals who use ML, teachers can better internalise the goals of the school and sustainable education policies. Indeed, the main element of 'the school principal's work is primarily verbal communication that is both interpersonal and informational' [72] (p. 858). If school leaders regularly adopt ML, they may even enhance teachers' commitment to the teaching profession. The recent study by Akça [24] also points to the fact that the beliefs of prospective teachers regarding education for sustainable development are influenced by their self-efficacy levels as well as ability to focus on solutions. Given this potential, the current study presents the analytical importance of school principals' language use in motivating teachers via cultural context and provides support for Skaalvik and Skaalvik's [25] findings. Motivating language use by school principals is crucial to enhance positive and sustainable school climate [44], offer clear work expectations for teachers particularly in high context cultures as well as to improve teachers' self-efficacy [73]. Teachers with high self-efficacy use appropriate techniques and exert extra effort in order to teach unmotivated students, whereas teachers with low or without any self-efficacy believe they can do little or nothing to teach unmotivated students [26]. Additionally, teachers' self-efficacy beliefs can facilitate handling disciplinary matters and ensure sustainable learning environment in the classroom [74]. Ball [75] suggested that teachers with high self-efficacy dedicate themselves to maintaining interactive, sustainable relationships with parents and colleagues, which in turn contribute stimulating and motivating learning contexts.

We can understand that teachers from low-context cultures are more inclined to open communication and are motivated by it, and they have higher self-efficacy beliefs compared to those from high-context cultures. This study also shows that school principals need to consider teachers' different cultural backgrounds and should not refrain from turning to implicit communication methods specific to Turkish culture while using open communication channels. As Moradkhani and Haghi [15] claimed, the potency of different sources of self-efficacy, such as verbal persuasion, can be a function of the significance that teachers of a particular culture attach to them. Thus, our findings in this study are also likely to challenge the assertion put forward by Bandura [76], namely, that self-efficacy beliefs work in a similar way across cultures.

On the other hand, it was also observed that individuals with high self-efficacy are less (but significantly) affected using ML (the impact factor was found to be as small as 0.16). One of the reasons for this effect might be the high intrinsic motivation of such individuals. Studies have confirmed that high-performing individuals are affected by intrinsic rather than extrinsic motivation [77], and self-efficacy is suggested to be one of the antecedents of intrinsic motivation [78]. There are even studies claiming that verbal motivation is not as effective on self-efficacy [39]. For example, a beginning teacher may be more affected by ML than an experienced teacher. If the high number of experienced teachers that participated in this study prevented us from making such a comparison, future studies should take this issue into consideration. One of the rare studies [39] that investigated this difference found existing 'teaching resources and interpersonal support' affected novice teachers' self-efficacy beliefs, whereas for the experienced teachers, mastery experiences seemed to be more important. Additionally, Bandura [23] himself posited that self-efficacy remains stable over time once acquired. Among Iranian teachers, however, verbal persuasion was identified as the most important source of self-efficacy, more than mastery experience [15], which is also in line with the major findings of our study. Furthermore, instead of measuring teacher self-efficacy with a general self-efficacy scale, it would be more fruitful to measure it with the educator-specific 'Teacher Self-Efficacy Scale'. Tschannen-Moran and Hoy [39] argued that different self-efficacy scales can provide inconsistent results since the teacher self-efficacy scale incorporates measures based on teaching-related self-efficacy beliefs. In a similar vein, as Bellibas and Liu [27] indicated, the current literature has largely neglected the multidimensional aspects of self-efficacy and approached it as a single and unitary construct.

The literature [39] regarding teacher self-efficacy found a strong relationship with 'many meaningful, sustainable educational outcomes such as teachers' persistence, enthusiasm, commitment, and instructional behaviour, as well as student outcomes such as achievement, motivation, and self-efficacy beliefs' (p. 783). On the other hand, the substantial evidence showing the power of motivational variables on teaching effectiveness is even greater than the effect of personality traits [79]. Thus, it is essential to keep searching for the factors that may increase teachers' perception of their self-efficacy beliefs.

Other antecedents of general self-efficacy of employees exist in the literature. For example, the causal relationship has been founded regarding the role of self-efficacy on motivating individuals. The other way of such relationship has been rarely subject to scholarly investigation. By referring to Lunenburg, ref. [80] argues that goal-setting theory and self-efficacy theory are complementing each other. Setting difficult goals for employees by their managers could lead them to have a higher level of self-efficacy. However, the issue of whether each employee reacts the same to difficult tasks and stays motivated is still questionable. Here, then, ML could be resourceful for managers when they are confronted with this type of case. If they realize that the employee is about to give up, they could use ML to encourage them to not to give up and continue.

Certain studies have underlined that training in ML could help leaders to yield benefits such as innovation, job performance, self-efficacy, job satisfaction, decision-making, and leader effectiveness along with sustainable organisational accomplishments [5] (p. 4). School principals should take these potential benefits into consideration and, in order to create a difference with the use of ML, as Holmes and Carr [5] further argued, they should make use of situational awareness. The context where cultural differences are important could be considered such a situation, and the current study found

some evidence regarding the mediating effect of cultural context on the relationship between school principals' use of ML and teacher self-efficacy.

Lastly, like other studies, this study is not exempt from limitations. The first limitation is data collection procedures. We have collected all the data from one source. The reason for that was the notion of understanding multiple realities that are socially constructed based on how the teachers perceive the reality of motivating language of their principals. According to Munhall [81] "perceptions are interpretations, and for most individuals, interpretations become their truth. Thus, perceptions are extremely powerful and influential in human thought and behaviour." (p. 606). Although we have indicated that CMB is controlled as much as possible both in the administration process and after post-hoc analyses, it would be appropriate to collect data from two different sources separately: Principals and teachers. Therefore, a multilevel theoretical model could provide different perspectives. Second, the convenience sampling method is not adequate to infer generalization. Future studies should consider randomized sampling techniques. As for the strengths of our study, we provide evidence that theories from culture, language, and communication disciplines could enrich educational management and leadership studies. Thus, by highlighting the importance of multi-disciplinary approaches to theoretical and our proposed research model, we set a preliminary example to encourage scholars for future studies in the field.

## 5. Practical Implications

Given the master–apprentice relationship underlying teacher training, the master's exemplary behaviours and the apprentice's imitation of these behaviours can be important for improving teachers' self-efficacy. For this reason, verbal motivation can adversely affect individuals from high-context cultures and critical feedback can be perceived as a negative phenomenon. Individuals in high-context cultures where collectivist tendencies are predominant generally act or behave according to societal perception—a practice that leads to lower self-confidence about their abilities. Since they seek to avoid committing faults in the public eye, such individuals will not try to do any more than enough. They tend to assess critics as threats and search for approval in high-context cultures. School principals can use illocutionary speech acts (interactional relations) to overcome the negative effects of communication on such individuals.

Additionally, people from a low-context culture could be perceived as ignorant, rude, or incompetent in a high-context culture due to the way they behave. For example, it is quite natural for them to ask a lot of questions (hence implying that they do not understand the meaning without them), or act in a confrontational way. Sometimes, they might not even know how to fit into the group dynamics or are unable to cope up with many tasks simultaneously. Likewise, an individual from a high-context culture could be considered vague, secretive, unpunctual, unable to adhere to plans, or incompetent due to a lack of ability to work on their own by the low-context culture [82]. School principals should be aware of such group dynamics as well as potential conflicts among teachers or between teachers and themselves.

## 6. Suggestions for Future Research

Future studies should examine the effect of implicit communication methods, together with ML, on teachers' self-efficacy beliefs in high-context cultures. The effects of low-context cultures on positive interactive language [58] were observed in this study. In order to reach more generalizable results, it is crucial to conduct and replicate similar research in high-context cultures. Future research in other contexts with high- and low-context cultures is also needed to formulate theoretically relevant and statistically meaningful generalisations in this regard.

Arslan and Yener [83] found in their study of paternalistic managers and employee individualism-collectivism (Among Hofstede's 5 D) relationship that employees with individualistic tendencies do not like paternalist features of their managers whereas the others find relatedness and closeness. Individualistic employees prefer to be explicit in their encounters, and this reminds us they

can fit best to low-context culture. As a further research avenue, it would be interesting to research how teachers from individualist and collectivist tendencies will respond to motivational language use by their principals.

Hofstede proposed that individuals who are prone to power distance and uncertainty avoidance talk less either because they fear their superiors or because talking might put them into troubles in which they could not know how to handle [84]. Instead, they choose symbols and gestures/mimics similarly as high-low context individuals would do. Previous studies have shown that in low power-distance cultures, organisational members consider less rich (lean) communication media more effective whereas organisational members in higher power-distance societies consider rich communication more effective [85]. In high power-distance cultures, managers are likely to communicate using different media than in lower-power-distance cultures and use them with high interactivity [85]. It would be subject to a future research to investigate the moderating influence of communication frequency and the choice of communication medium on the relationship proposed in the current study. Lastly, we did not consider whether the participants worked at the same schools. Clusters might affect the outcomes. It would be beneficial to investigate the school-level effects through multilevel models.

## 7. Conclusions

This study adds to the current understanding of the relationship between teachers' perception of school principals' motivating language and teachers' self-efficacy mediated by the cultural context. The results revealed the presence of a significant, positive effect of ML on low-context culture but a negative effect on high-context culture. Additionally, the low-context culture acted as a full mediator in this relationship. On the other hand, the negative effect might indicate the need to consider the previously mentioned different environmental factors and cues (style, gestures, mimics, body language, and stress) that describe the embedded and implicit communication in the Turkish cultural context.

Lastly, the main implication of this study is that specific policies should be developed to enhance teachers' self-efficacy through training school principals how to use effectively motivating language use and engage teachers with several ways to increase their personal and professional growth [86].

**Author Contributions:** Conceptualization and study design, E.O.; data collection, E.O. and A.A. (Aykut Arslan); data analysis and methodology, A.A. (Aykut Arslan) and S.Y.; software and validation, A.A. (Aykut Arslan) and S.Y.; interpretation and discussion of results, E.O. A.A. (Andrea Appolloni); Writing—Original draft preparation, E.O., A.A. (Aykut Arslan) and A.A. (Andrea Appolloni); Writing—Review and editing, E.O. and A.A. (Andrea Appolloni); visualization, S.Y.; project administration, E.O. and A.A. (Andrea Appolloni), founding A.A. (Andrea Appolloni). All authors have read and agreed to the published version of the manuscript.

**Funding:** This research received external funding from European Project Marie-Curie Horizon 2020.

**Conflicts of Interest:** The authors declare no conflict of interest.

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
