# Peer review of "The Predictive Effect of Teachers’ Perception of School Principals’ Motivating Language on Teachers’ Self-Efficacy via a Cultural Context"

_sustainability, doi:10.3390/su12218830_

Round 1

Reviewer 1 Report

I want to thank you for the opportunity to review the article entitled: “The Effect of School Principals' Use of Motivating Language on Teachers' Self-Efficacy via a Sustainable Cultural Context”

Grounded in Motivating language theory (MLT), this study aimed to examine the mediating effect of cultural context on the relationship between teachers' perception of school principals’ motivating language and self-efficacy. This cross-sectional study was conducted using a small convenience sample of teachers. Therefore, it was not possible to test the direction of causality between variables. Moreover, self-reported measures were used to collect data.

The entire manuscript needs significant work to improve the language, grammar and sentence structure. There are several issues with tenses throughout. I understand that English may not be the first language of the author/s and that although essential that it is improved, it is an aspect that can be worked on. The errors throughout are distracting to the flow of the paper in its current form. The correction to the writing style may also serve to streamline many of the sections. If a revision is recommended, the authors are highly encouraged to seek assistance with editing grammatical errors within the manuscript.

Therefore, although the topic is interesting, the concurrent design and exclusive reliance on self-report measures are important limitations. Based on my reading of your manuscript, I have decided that this manuscript is unlikely to be publish in a prestigious journal as Sustainability.

I detail my concerns below.

Tittle

  • The title is somewhat confusing because it seems that an intervention programme has been carried out.
  • I would put “predictive effect”.
  • What does mean “Sustainable Cultural Context” I would put “Cultural Context”.
  • The name of the variable “School Principals' Use of Motivating Language” is confusing as it seems to have been measured on principals rather than teachers. I would put “teachers' perception of school principals’ motivating language”

Abstract

  • The objective of the study is not clearly detailed.
  • The study design and type of sampling should be detailed in the abstract.
  • Please, add the average age and the standard deviation.
  • Why do the authors point out “The psychometric properties of all three scales were established” Were the questionnaires not validated? Which questionnaires do you refer to?
  • The name of the variable “School Principals' Use of Motivating Language” is confusing as it seems to have been measured on principals rather than teachers. I would put “teachers' perception of school principals’ motivating language”
  • Author said “The results showed that school principals’ use of motivating language was effective in teachers' self-efficacy (b=0.10, p< 23 .000)” This sentence is somewhat confusing because it seems that an intervention programme has been carried out. I would use words such as “…was significantly and positively associated with…”
  • The p's are in italics
  • I would remove this keyword “low and high context culture” because it is a very long term.

Introduction

  • The introduction is very long. The authors have to synthesize all the information highlighting the state of the art. In the paragraphs and pages that follow, the authors need to identify pertinent literatura and explain how their study builds upon and extends such literature. The Introduction of an empirical paper is not a section where one needs to exhaustedly summarize “everything” that one knows about a topic. An exhaustive examination of a topic would be more appropriate for a review paper. Rather, the purpose of the Introduction section in relation to an empirical study is to present the reader to the literatura that is most pertinent in relation to the manuscript If there has already been a lot of research on the specific topic, the included studies should ideally come from samples or research participants that share similar characteristics to the manuscript’s sample.
  • The numbers of each reference are not well placed at the end of sentences. For example, [8] [1-2]; [5] [2].
  • Line 55, I would put “Motivating language theory (MTL)”.
  • I suggest that the opening paragraph of this manuscript was not so extensive as all the information is explained later.
  • I would not divide the self-efficacy paragraph into two headings.
  • Authors said “Studies show that teacher self-efficacy can be related to and predict teachers’ different cognitive, emotional, and behavioural responses; for example, it is correlated positively with teacher job satisfaction and job commitment, but negatively with burnout and teacher attrition [23] (p.101). High teacher self-efficacy is considered to potentially lead to positive student outcomes [17]; [24]; [25]. Teachers with high self-efficacy are able to plan effective instructional strategies, increase sustainable performance, and enhance their effectiveness and productivity [26] (p.79). These teachers can also help students with academic adjustments as well as contribute to sustainable classroom quality [27]. Moreover, teachers with high self-efficacy are important in the adaptation and upgrading of the education process according to the changing needs of the era [28].” However, I would suggest that the authors support these sentences with systematic reviews or meta-analyses.
  •  
  • Chesnut, S. R., & Burley, H. (2015). Self-efficacy as a predictor of commitment to the teaching profession: A meta-analysis. Educational Research Review15, 1-16.
  • Aloe, A. M., Amo, L. C., & Shanahan, M. E. (2014). Classroom management self-efficacy and burnout: A multivariate meta-analysis. Educational psychology review26(1), 101-126.
  • Klassen, R. M., & Tze, V. M. (2014). Teachers’ self-efficacy, personality, and teaching effectiveness: A meta-analysis. Educational Research Review12, 59-76.
  • Shoji, K., Cieslak, R., Smoktunowicz, E., Rogala, A., Benight, C. C., & Luszczynska, A. (2016). Associations between job burnout and self-efficacy: a meta-analysis. Anxiety, Stress, & Coping29(4), 367-386.
  • Kim, K. R., & Seo, E. H. (2018). The relationship between teacher efficacy and students' academic achievement: A meta-analysis. Social Behavior and Personality: an international journal46(4), 529-540.
  • In this section “2.4. Relationship between Motivating Language and Self-Efficacy” I would recommend that the authors focus on the relationship between these two variables in teacher samples. There is a lot of information that could be omitted and moves the focus away from the study. For example, I would remove this study becaused was conducted in a sample of soldiers. “The experimental study Eden & Kinnar [31] conducted on Israeli soldiers reported that verbal persuasion increased the self-efficacy of soldiers in the Israeli Special Forces and led to a considerable increase in their rates of volunteering” Moreover, i would remove the following sentences because in the previous section, you explain the relatiosnhip between self-efficacy and other (mal)aladaptive outcomes “A study conducted with over 2,000 teachers from 75 different secondary schools in Italy [33] revealed that teachers’ self-efficacy beliefs affected their job satisfaction as well as their students’ academic performance. A review of the literature from the past three decades shows that various studies have confirmed the presence of a direct and positive relationship between teachers’ self-efficacy and students’ self-efficacy, motivation, and academic performance [34].”
  • The authors identify the first hypothesis of the study without explaining the objective of this study.
  • Authors said “Another important factor worth considering in the relationship between the use of ML by school principals and teachers’ self-efficacy is cultural context” You need to provide a reference to support this claim.

Method

  • Authors should add several headings before “measures” section. For example, “design”, “participants”, “context”, etc.
  • The section of the method is not well sequenced. For example, the data analysis is explained in the second sentence. I suggest the authors put a specific heading to explain “data analysis” after the heading of “measures”.
  • It would be useful to explain the context where the study was conducted in order to understand the role of principals.
  • Please, add the average age, standard deviation, teacher experience, type of schools, etc. More information on the socio-demographic variables of the sample should be added.
  • The measurement section is very schematic and the factors that make up each variable are not known. I suggest the authors add the names and number of items of the factors of each instrument and an example for better understanding.

For example,

Teachers' need-supportive behaviours. Students' perceptions of autonomy, competence, and relatedness support from PE teacher were assesed by the Spanish version of the Questionnaire of Basic Psychological Needs Support in Physical Education (Sánchez-Oliva, Leo, Amado, Cuevas, & García-Calvo, 2013). The statement “In PE classes, my teacher…” was followed by 12 items (four items per factor) that assess: autonomy support (e.g., “Often asks us about our preferences with respect to the activities we carry out”), competence support (e.g., “Offers us activities based on our skill level”), and relatedness support (e.g., “Encourages positive interactions among all pupils”). Responses were recorded on 5-point scale ranging from 1 (strongly disagree) to 5 (strongly agree).

Discussion

  • After discussing all their findings, authors should summarize the major strengths and limitations of their work, with reference to (where relevant) conceptual development or refinement, methodology, and relevance for practice and policy. Embedded in outlining the major strengths and limitations of their work, we also suggest that authors provide a small number of recommendations for future research. Such avenues for future research can draw directly or indirectly from the manuscript’s findings or limitations; authors are advised to explain why such future avenues for research are important.

Conclusions

  • The conclusions section is very extensive. The conclusion section is a form of summary, or synthesis, derived from the project. There are meant to be clear takeaways from the project, and these need to be highlighted, in order to reinforce one’s message. These “conclusions” should be drawn directly from the study’s findings, logically, and articulate the most important lessons and insights. Conclusions are also meant to be concise, meaning that they should be short and powerful, emphasizing what the reader should readily extract from the project. This is the author’s final opportunity to leave a lasting impression, one that might strengthen one’s argument that the submission should proceed to revision or, in some cases, outright acceptance.

Author Response

Response to Reviewer 1 Comments

Point 1: This cross-sectional study was conducted using a small convenience sample of teachers. Therefore, it was not possible to test the direction of causality between variables. Moreover, self-reported measures were used to collect data. Therefore, although the topic is interesting, the concurrent design and exclusive reliance on self-report measures are important limitations.

Response 1: The reviewer raised a very relevant issue addressing implicitly common method variance derived from the concurrent design and exclusive reliance on self-report measures. In line with such an insightful comment by the reviewer, we opened and added a separate part under method section (please see 3.3.1. Common Method Variance).  Here we discussed briefly the procedural steps followed in the study in order to minimize common method variance as much as possible both in the administration process and after post-hoc analyses. The added part (line 363-374) in the revised manuscript was shown below:

“To reduce the possibility of common method bias (CMB) we follow three procedural remedies. First, we shared the self-efficacy and motivational language scales. Then, after 4 weeks we collected cultural context variables [73]. This temporal, psychological or methodological separation of measured items/measurement of variables is suitable to reduce CMB when the researcher is unable to collect data from multiple sources (p.146). Second, we checked the VIFs and tolerance values. These were below 3 and the tolerance values were below 1 (see Table 4), indicating the absence of CMB [74]. Lastly, we run a Harman’s Single-Factor Test. In this test all items (measuring latent variables) are loaded into one common factor. It is expected that the total variance for a single factor should be less than 50%; our test results indicated a total variance of 41, 95%. Thus, we conclude that CMB was minimized as much as possible both in the administration process and after post-hoc analyses”.

Point 2: The entire manuscript needs significant work to improve the language, grammar and sentence structure. There are several issues with tenses throughout. I understand that English may not be the first language of the author/s and that although essential that it is improved, it is an aspect that can be worked on. The errors throughout are distracting to the flow of the paper in its current form. The correction to the writing style may also serve to streamline many of the sections. If a revision is recommended, the authors are highly encouraged to seek assistance with editing grammatical errors within the manuscript.

Response 2: The language of the entire manuscript has been rechecked and edited by a professional proof-reader. All the issues regarding grammar, sentence structure and typing errors have been clearly highlighted. The authors have made significant corrections based on the proof-reader’s recommendations. We hope that the flow of the revised version has been greatly improved.

Point 3:

Title

  • The title is somewhat confusing because it seems that an intervention programme has been carried out. I would put “predictive effect”.
  • What does mean “Sustainable Cultural Context” I would put “Cultural Context”.
  • The name of the variable “School Principals' Use of Motivating Language” is confusing as it seems to have been measured on principals rather than teachers. I would put “teachers' perception of school principals’ motivating language”

Abstract

  • The objective of the study is not clearly detailed.
  • The study design and type of sampling should be detailed in the abstract. Please, add the average age and the standard deviation.
  • Why do the authors point out “The psychometric properties of all three scales were established” Were the questionnaires not validated? Which questionnaires do you refer to?
  • The name of the variable “School Principals' Use of Motivating Language” is confusing as it seems to have been measured on principals rather than teachers. I would put “teachers' perception of school principals’ motivating language”
  • Author said “The results showed that school principals’ use of motivating language was effective in teachers' self-efficacy (b=0.10, p< 23 .000)” This sentence is somewhat confusing because it seems that an intervention programme has been carried out. I would use words such as “…was significantly and positively associated with…”
  • The p's are in italics. I would remove this keyword “low and high context culture” because it is a very long term.

Response 3:

Title

  • The title has been rearranged and the new title in the revised manuscript is shown below:

“The Predictive Effect of Teachers’ Perception of School Principals' Motivating Language on Teachers' Self-Efficacy via a Cultural Context”

  • The sustainable cultural context has been removed from the title and ‘cultural context’ has remained in the title.
  • The name of the variable “School Principals' Use of Motivating Language” has been replaced with “teachers' perception of school principals’ motivating language” in order to eliminate the confusion addressed by the reviewer.

Abstract

  • The objective of the study has been added to the very beginning of the abstract as can be seen in the following (line 16-17):

“The aim of the study is to investigate the relationship between teachers’ perception of school principals’ motivating language and teachers’ self-efficacy mediated by the cultural context”.

  • The study design (survey method) and type of sampling (convenience sampling) were addressed in details. Moreover, average age and standard deviation were added. Please check the following sentences (line 21-23):

“A survey method was employed with 252 teachers through convenience sampling. The teachers’ mean age was 34.87 [SD = 9.22] years, and the average length of service was 11.72 [SD = 9.42] years”.

  • The reviewer raised a very relevant issue and we have decided to not to mention about the psychometric properties of scales adopted in the study because they have been validated previously in the Turkish context. Thus, we removed this sentence from the abstract: “The psychometric properties of all three scales were established” Were the questionnaires not validated?”

  • We have now included the statement of “teachers' perception of school principals’ motivating language” rather than “School Principals' Use of Motivating Language” not only in the title but also in the abstract (please see line 16)

  • We have now used the words as “…was significantly and positively associated with…” in order to eliminate such confusion addressed by the reviewer. Please see the revised sentence in the abstract (line 23-25):

“The results showed that school principals’ use of motivating language was significantly and positively associated with teachers' self-efficacy (b=0.10, p< .000)”.

  • In line with the reviewer’s suggestion, the p's are now in italics and we have removed the keyword “low and high context culture” as it is too long for the reviewer.

Point 4:

Introduction

The introduction is very long. The authors have to synthesize all the information highlighting the state of the art. In the paragraphs and pages that follow, the authors need to identify pertinent literature and explain how their study builds upon and extends such literature. The Introduction of an empirical paper is not a section where one needs to exhaustedly summarize “everything” that one knows about a topic. An exhaustive examination of a topic would be more appropriate for a review paper. Rather, the purpose of the Introduction section in relation to an empirical study is to present the reader to the literature that is most pertinent in relation to the manuscript If there has already been a lot of research on the specific topic, the included studies should ideally come from samples or research participants that share similar characteristics to the manuscript’s sample.

  • The numbers of each reference are not well placed at the end of sentences. For example, [8] [1-2]; [5] [2].
  • Line 55, I would put “Motivating language theory (MTL)”.
  • I suggest that the opening paragraph of this manuscript was not so extensive as all the information is explained later. I would not divide the self-efficacy paragraph into two headings.
  • Authors said “Studies show that teacher self-efficacy can be related to and predict teachers’ different cognitive, emotional, and behavioural responses; for example, it is correlated positively with teacher job satisfaction and job commitment, but negatively with burnout and teacher attrition [23] (p.101). High teacher self-efficacy is considered to potentially lead to positive student outcomes [17]; [24]; [25]. Teachers with high self-efficacy are able to plan effective instructional strategies, increase sustainable performance, and enhance their effectiveness and productivity [26] (p.79). These teachers can also help students with academic adjustments as well as contribute to sustainable classroom quality [27]. Moreover, teachers with high self-efficacy are important in the adaptation and upgrading of the education process according to the changing needs of the era [28].” However, I would suggest that the authors support these sentences with systematic reviews or meta-analyses.

Chesnut, S. R., & Burley, H. (2015). Self-efficacy as a predictor of commitment to the teaching profession: A meta-analysis. Educational Research Review, 15, 1-16.

Aloe, A. M., Amo, L. C., & Shanahan, M. E. (2014). Classroom management self-efficacy and burnout: A multivariate meta-analysis. Educational psychology review, 26(1), 101-126.

Klassen, R. M., & Tze, V. M. (2014). Teachers’ self-efficacy, personality, and teaching effectiveness: A meta-analysis. Educational Research Review, 12, 59-76.

Shoji, K., Cieslak, R., Smoktunowicz, E., Rogala, A., Benight, C. C., & Luszczynska, A. (2016). Associations between job burnout and self-efficacy: a meta-analysis. Anxiety, Stress, & Coping, 29(4), 367-386.

Kim, K. R., & Seo, E. H. (2018). The relationship between teacher efficacy and students' academic achievement: A meta-analysis. Social Behavior and Personality: an international journal, 46(4), 529-540.

  • In this section “2.4. Relationship between Motivating Language and Self-Efficacy” I would recommend that the authors focus on the relationship between these two variables in teacher samples. There is a lot of information that could be omitted and moves the focus away from the study. For example, I would remove this study because was conducted in a sample of soldiers. “The experimental study Eden & Kinnar [31] conducted on Israeli soldiers reported that verbal persuasion increased the self-efficacy of soldiers in the Israeli Special Forces and led to a considerable increase in their rates of volunteering” Moreover, I would remove the following sentences because in the previous section, you explain the relationship between self-efficacy and other (mal)adaptive outcomes “A study conducted with over 2,000 teachers from 75 different secondary schools in Italy [33] revealed that teachers’ self-efficacy beliefs affected their job satisfaction as well as their students’ academic performance. A review of the literature from the past three decades shows that various studies have confirmed the presence of a direct and positive relationship between teachers’ self-efficacy and students’ self-efficacy, motivation, and academic performance [34].”
  • The authors identify the first hypothesis of the study without explaining the objective of this study.
  • Authors said “Another important factor worth considering in the relationship between the use of ML by school principals and teachers’ self-efficacy is cultural context” You need to provide a reference to support this claim

Response 4:

Introduction

The introduction section has been rearranged to make it succinct for the reader by highlighting the state of the art literature, major theoretical background adopted in the study as well as the aim, research gap and key contribution. The overall structure of the paper is presented in the very end of introduction.

  • Please kindly check line 75-99 for the newly added part in the introduction and see below the statements used in the revised manuscript:

“Despite the fact that a significant body of literature and discussion surrounding education for sustainable development exists, the language and linguistic communication skills school principals have to sustain the motivation of teachers and enhance their self-efficacy has received scant attention. In our study, we partially address this gap in the literature by clearly highlighting the aspects of motivational language use by school principals to enhance teachers’ self-efficacy in the light of cultural context. School principals’ influence on teachers through motivational language use is substantial as the level of teachers’ self-efficacy increases, teachers’ burn out and attrition can be lessened and hence, this may result in more motivated, committed and sustained workforce of teachers as well as higher students’ academic achievement [17,18] and teaching effectiveness [19] that are all likely to contribute to education for sustainable development. While education for sustainable development enables people to understand the influences of their own actions by encouraging them to take responsible decision [20], enhancing teachers’ self-efficacy through motivational language use by school principals can serve this purpose as teachers with high self-efficacy hold the sense of being able to achieve a certain task in a specific situation. More specifically, teachers with high self-efficacy believe that they can teach even the most difficult and unmotivated students so that they can improve student achievement [21].

The paper is organised as follows: The second section presents the literature review in six subheadings that are ‘motivating language theory’, ‘self-efficacy in teaching’, ‘relationship between motivating language and self-efficacy’, ‘cultural context’, ‘relationship between motivating language and cultural context’ and ‘the mediating role of cultural context’. The third section explains the methodology part in greater details including research design, sample and data analysis (‘common method variance’, ‘measures’, ‘validation of the measures’, ‘results of hypothesis tests’). The fourth section discusses the results in the light of literature. The fifth section presents practical implications derived from this study. The sixth section offers suggestions for future research. The last section presents the concluding remarks of the manuscript”.

  • The numbers of each reference are now well placed at the end of sentences throughout the manuscript.
  • We put “Motivating language theory (MLT)” in the revised submission (please see line 51).

  • In line with reviewer’s comment, the introduction section has been extended to give a clear idea and focus of the study to the reader (please see the added parts in line 75-99). Self-efficacy paragraph has been combined and renamed as only one heading “Self-Efficacy in Teaching” (please see line 130) in the revised manuscript.
  • In line with the reviewer’s suggested sources, we have now supported these sentences by incorporating several ideas and evidence based on systematic reviews or meta-analyses. Please kindly check line 163-169 for newly added parts to support this claim and also see below the statements used in the revised manuscript:

“There is a large body of evidence supported by multivariate meta-analyses on teachers’ self-efficacy. Accordingly, classroom management self-efficacy of teachers was significantly and negatively associated with burnout [21; 37] and positively associated with students’ academic achievement (17). Teachers’ self-efficacy was not only found to enhance students’ academic achievement but also overall teaching effectiveness (19). In predicting commitment to the teaching profession of preservice and inservice teachers, their self-efficacy beliefs can be utilized [38]”.

  • In the section of “2.3. Relationship between Motivating Language and Self-Efficacy”, we have focused particularly on the relationship between these two variables in teacher samples by omitting several studies reviewer 1 has found irrelevant for the study.
  • We have now identified the first hypothesis of the study with clearly explaining the objective of this study. Please kindly check line 208-210 for the newly added study’s objective statement to support this claim in the revised manuscript:

“Based on the objective of this study that is to examine the effect of teachers’ perception of school principals’ motivating language on teachers’ self-efficacy via their cultural contexts along with aforementioned robust evidence in the literature, the first hypothesis of this study is as follows”.

  • We have provided proper references to support the role of cultural context in the relationship between teachers’ perceptions of school principals’ motivating language and teachers’ self-efficacy. Please kindly check line 219 for newly added citations [50,51,52,53].

Point 5:

Method

  • Authors should add several headings before “measures” section. For example, “design”, “participants”, “context”, etc. The section of the method is not well sequenced. For example, the data analysis is explained in the second sentence. I suggest the authors put a specific heading to explain “data analysis” after the heading of “measures”.

  • It would be useful to explain the context where the study was conducted in order to understand the role of principals.

  • Please, add the average age, standard deviation, teacher experience, type of schools, etc. More information on the socio-demographic variables of the sample should be added.

  • The measurement section is very schematic and the factors that make up each variable are not known. I suggest the authors add the names and number of items of the factors of each instrument and an example for better understanding.

For example,

Teachers' need-supportive behaviours. Students' perceptions of autonomy, competence, and relatedness support from PE teacher were assesed by the Spanish version of the Questionnaire of Basic Psychological Needs Support in Physical Education (Sánchez-Oliva, Leo, Amado, Cuevas, & García-Calvo, 2013). The statement “In PE classes, my teacher…” was followed by 12 items (four items per factor) that assess: autonomy support (e.g., “Often asks us about our preferences with respect to the activities we carry out”), competence support (e.g., “Offers us activities based on our skill level”), and relatedness support (e.g., “Encourages positive interactions among all pupils”). Responses were recorded on 5-point scale ranging from 1 (strongly disagree) to 5 (strongly agree).

Response 5:

  1. We have added several new headings before “measures” section such as “Design”, “Sample”, “Data Analyses”. We put a specific heading to explain data analysis after the heading of measures in line with the reviewer’s suggestion. We have reviewed and rearranged completely the method section. Accordingly, the sequence and structure of the method section in the revised manuscript is kindly presented below:
  2. Method

3.1. Design

3.2. Sample

3.3. Data Analyses

3.3.1. Common Method Variance

3.3.2. Measures

3.3.2.1. Validation of the Measure

3.3.3. Results of Hypothesis Tests

  1. We have added the following statements to the end of 3.2 Sample section (line 358-360) to explain the context where the study was conducted in order to understand the role of principals.

“The study took place in public schools only. The number of teachers working at elementary level schools [grades 1-4] were 48 (19%); at secondary level schools [grades 5-8] were 84 (33%), and at high school level [grades 9-12] were 120 (48%)”.

  1. We have included the average age, standard deviation; teacher experience and type of school in section 3.2 Sample (line 352-360). Additionally, more information on the socio-demographic variables of the sample were added and shown in Table 5 (line 361).

  1. Thanks to Reviewer 1 for providing us such a clear example. Based on this direction, we have added the names and number of items of the factors of each instrument. Please kindly check section 3.3.2. Measures (line 375) for the revised parts. It is shown below:

“All statements were rated on a five-point Likert scale ranging from 1 (strongly disagree) to 5 (strongly agree).

Motivating language scale: The scale was developed by Mayfield & Mayfield [69] and adapted into Turkish by Özen [48]. It comprises three factors and 24 questions. The following is a sample item: “Provides me with easily understandable instructions about my work. “

General self-efficacy scale: The scale was developed by Sherer et al [70] and adapted into Turkish by Yıldırım & İlhan [61]. The following is a sample item: ‘I can usually handle whatever comes my way’.

Cultural communication scale: The scale was developed by Erdem [62] in accordance with Turkish culture largely based on Hall’s model. The following is a sample item: ‘While communicating in our society, one should pay attention to what is implied but not to what is said’.

The mean scores of latent variables were used in the study; thus, factors were irrelevant for the analyses. However, it is advised to report how many factors are extracted and see if the scales fit to their originals with this regard”.

Point 6:

Discussion

After discussing all their findings, authors should summarize the major strengths and limitations of their work, with reference to (where relevant) conceptual development or refinement, methodology, and relevance for practice and policy. Embedded in outlining the major strengths and limitations of their work, we also suggest that authors provide a small number of recommendations for future research. Such avenues for future research can draw directly or indirectly from the manuscript’s findings or limitations; authors are advised to explain why such future avenues for research are important.

Conclusions

The conclusions section is very extensive. The conclusion section is a form of summary, or synthesis, derived from the project. There are meant to be clear takeaways from the project, and these need to be highlighted, in order to reinforce one’s message. These “conclusions” should be drawn directly from the study’s findings, logically, and articulate the most important lessons and insights. Conclusions are also meant to be concise, meaning that they should be short and powerful, emphasizing what the reader should readily extract from the project. This is the author’s final opportunity to leave a lasting impression, one that might strengthen one’s argument that the submission should proceed to revision or, in some cases, outright acceptance.

Response 6:

Discussion

The discussion section has been significantly improved, rearranged and extended in line with the reviewer’s suggestion of including limitations of the work, relevance for practice and policy as well as a number of recommendations for future research. We drew future research avenues directly or indirectly from the current study’s findings and limitations and explained in greater details why such future research avenues are important.

  • Please kindly check line 523-531 for the newly added part in the revised manuscript:

“Other antecedents of general self-efficacy of employees exist in the literature. For example, the causal relationship has been founded regarding the role of self-efficacy on motivating individuals. The other way of such relationship has been rarely subject to scholarly investigation. By referring to Lunenburg [80] argues that goal-setting theory and self-efficacy theory are complementing each other. Setting difficult goals for employees by their managers could lead them to have a higher level of self-efficacy. But the issue whether each employee react the same to difficult tasks and motivated is still questionable. Here, then, ML could be resourceful for managers when they confront with this type of case. If they realize that the employee is about to give up, they could use ML to encourage them to not to give up and continue”.

  • Please kindly check line 540-554 for the newly added part in the revised manuscript:

“Lastly, likewise other studies, this study is not exempt from limitations. The first limitation is data collection procedures. We have collected all the data from one source. The reason for that was the notion of understanding multiple realities that are socially constructed based on how the teachers perceive the reality of motivating language of their principals. According to Munhall [81] “perceptions are interpretations, and for most individuals, interpretations become their truth. Thus, perceptions are extremely powerful and influential in human thought and behaviour.” (p.606) Although we have indicated that CMB is controlled as much as possible both in the administration process and after post-hoc analyses, it would be appropriate to collect data from two different sources separately: principals and teachers. Therefore, a multilevel theoretical model could provide different perspectives. Second, the convenience sampling method is not adequate to infer generalization. Future studies should consider randomized sampling techniques. As for the strengths of our study, we provide evidence that theories from culture, language and communication disciplines could enrich educational management and leadership studies. Thus, by highlighting the importance of multi-disciplinary approaches to theoretical and our proposed research model, we set a preliminary example to encourage scholars for future studies in the field”.

  • Please kindly check line 575-601 for the newly added, separate Section 6 Suggestions for Future Research in the revised manuscript:
  1. Suggestions for Future Research

Future studies should examine the effect of implicit communication methods, together with ML, on teachers’ self-efficacy beliefs in high-context cultures. The effects of low-context cultures on positive interactive language [58] were observed in this study. In order to reach more generalizable results, it is crucial to conduct and replicate similar research in high-context cultures. Future research in other contexts with high- and low-context cultures is also needed to formulate theoretically relevant and statistically meaningful generalisations in this regard.

Arslan and Yener [83] found in their study of paternalistic managers and employee individualism-collectivism (Among Hofstede’s 5 D) relationship that employees with individualistic tendencies do not like paternalist features of their managers whereas the others find relatedness and closeness. Individualistic employees prefer to be explicit in their encounters, and this reminds us they can fit best to low-context culture. As a further research avenue, it would be interesting to research on how teachers from individualist and collectivist tendencies will respond to motivational language use by their principals.

Hofstede proposed that individuals who are prone to power distance and uncertainty avoidance talk less either because they fear their superiors or because talking might put them into troubles in which they could not know how to handle [84]. Instead, they choose symbols and gestures/mimics similarly as high-low context individuals would do. Previous studies have shown that in low power-distance cultures, organisational members consider less rich (lean) communication media more effective whereas organisational members in higher power-distance societies consider rich communication more effective [85]. In high power-distance cultures, managers are likely to communicate using different media than in lower-power-distance cultures and use them with high interactivity [85]. It would be subject to a future research to investigate the moderating influence of communication frequency and the choice of communication medium on the relationship proposed in the current study. Lastly, we did not consider whether the participants worked at the same schools. Clusters might affect the outcomes. It would be beneficial to investigate the school-level effects through multilevel models.

  • We have also strengthened and extended the practical implications. Please kindly check line 566-574 for the newly added paragraph to Section 5 Practical Implications in the revised manuscript:

“Additionally, people from a low-context culture could be perceived as ignorant, rude, or incompetent in a high-context culture due to the way they behave. For example, it is quite natural for them to ask a lot of questions (hence implying that they do not understand the meaning without them), or act in a confrontational way. Sometimes, they might not even know how to fit into the group dynamics or are unable to cope up with many tasks simultaneously. Likewise, an individual from a high-context culture could be considered vague, secretive, unpunctual, unable to adhere to plans, or incompetent due to a lack of ability to work on their own by the low-context culture [82]. School principals should be aware of such group dynamics as well as potential conflicts among teachers or between teachers and themselves”.

Conclusions

In line with the reviewer’s suggestion, the conclusion section has been rearranged to a great extent to make it succinct, ‘to the point’ that a reader can easily derive clear takeaways. Please kindly check line 602-614 for the shortened version of conclusion section and see below the statements used in the revised manuscript.

“This study adds to the current understanding of the relationship between teachers’ perception of school principals’ motivating language and teachers’ self-efficacy mediated by the cultural context. The results revealed the presence of a significant, positive effect of ML on low-context culture but a negative effect on high-context culture. Additionally, the low-context culture acted as a full mediator in this relationship. On the other hand, the negative effect might indicate the need to consider the previously mentioned different environmental factors and cues [style, gestures, mimics, body language, and stress] that describe the embedded and implicit communication in the Turkish cultural context.

Lastly, the main implication of this study is that specific policies should be developed to enhance teachers’ self-efficacy through training school principals how to use effectively motivating language use and engage teachers with several ways to increase their personal and professional growth [86].”

Reviewer 2 Report

Review of the manuscript ‘The Effect of School Principals' Use of Motivating Language on Teachers' Self-Efficacy via a Sustainable Cultural Context´ (sustainability)

This manuscript has an interesting and well-grounded research frame focusing on the influence of the Motivation Language (ML) on Self-Efficacy with the mediating role of cultural context. The analysis is conducted and explained well, and the article is well-written. The focus is important and outcomes produce much needed knowledge on the linkage between ML and Self-Efficacy. Nevertheless, I think there are major considerations that need to be addressed before the manuscript can be accepted or even evaluated for the acceptance.

My biggest notion concerns the definitions and operationalizations of the ‘sustainable cultural context’ or generally the sustainability of the research frame. The viewpoint of sustainability and the theoretical conceptualizations seem to be missing from the manuscript. Sustainability is quite vaguely connected to ‘sustainable education settings’ (abstract), ‘communication’ (p. 1), ‘sustainable results’ / ‘performance’ / ‘mechanism’ / ‘overall sustainable development’ (p.2), among others, but these terms or the idea how the sustainability as a theoretical or analytical tool is used in the article is undefined. Before the overall evaluation of the manuscript, I think this needs to be clarified, for instance with the definitions offered by Education for Sustainable Development Goals (ESDG) and theoretical and empirical discussion on the special issue "Sustainability in Leadership and Education" in

The theoretical background offers yet quite coherent and compact discussions on the Motivation Language Theory, Self-Efficacy and the Cultural Context, as a narrow but well-defined theory of Hall’s low-context and high-context cultures. It might be beneficial for the reader to consider some other labels for the ‘low-context’ and ‘high-context’ culture in the results and especially in the outcomes and the abstract, since these labels are not really informative for the reader as such, without the theoretical explanation. Are there any alternative labels for these indicators?

The methodology of the research is well-defined and methods used properly. Outcomes are clearly stated. Nevertheless, it is difficult to understand how the principals were included in the data. As far as I can understand, the data was collected from the teachers, not the principals. How is the motivation language of the principals investigated? From the teachers? I find it really difficult to read this from the manuscript, since the data collection and the sample are narrowly explained. For instance, if the Motivational language of the principals were asked from the teachers (with the ML scale), why the questions were addressed to the teachers (e.g. the given example ‘I clarify what behaviours are expected at work’). Please, explain the data collection more thoroughly. As it is, I think that the causal impact of the ML on the self-efficacy is difficult to prove. I think this needs to be clarified and discussed in the research also more generally.

Finally, I encourage the authors to broaden the description of the convenience sampling of the study. The description is so scarce, that the nature and context of the data are not evident. The sample (teachers) are described before the analysis (section 4, results), but how about the schools they and the principals were working? For the generalizations it would be important to describe the schools in more detail (e.g. how many teacher worked in the public / private sectors, app. how large to schools were, how many of them were elementary, secondary or high schools). For the outcomes, it would be important to know if the participants worked at the same schools. Clusters might affect the outcomes. Can you estimate the school-level effects with the multilevel models?   

In all, I think this manuscript has potential, but there are major changes needed. The sustainability should be defined and included more coherently into the research frame, the information on the principals’ motivational language use should be explained more thoroughly and the sampling should be described in more detail.  

Author Response

Response to Reviewer 2 Comments

Point 1: This manuscript has an interesting and well-grounded research frame focusing on the influence of the Motivation Language (ML) on Self-Efficacy with the mediating role of cultural context. The analysis is conducted and explained well, and the article is well-written. The focus is important and outcomes produce much needed knowledge on the linkage between ML and Self-Efficacy.

Response 1: We would like to thank Reviewer 2 for overall evaluation about our study.

Point 2: My biggest notion concerns the definitions and operationalizations of the ‘sustainable cultural context’ or generally the sustainability of the research frame. The viewpoint of sustainability and the theoretical conceptualizations seem to be missing from the manuscript. Sustainability is quite vaguely connected to ‘sustainable education settings’ (abstract), ‘communication’ (p. 1), ‘sustainable results’ / ‘performance’ / ‘mechanism’ / ‘overall sustainable development’ (p.2), among others, but these terms or the idea how the sustainability as a theoretical or analytical tool is used in the article is undefined. Before the overall evaluation of the manuscript, I think this needs to be clarified, for instance with the definitions offered by Education for Sustainable Development Goals (ESDG) and theoretical and empirical discussion on the special issue "Sustainability in Leadership and Education".

The sustainability should be defined and included more coherently into the research frame.

Response 2: Based on the reviewer’s comment and insightful suggestion, the sustainability aspect has been incorporated to the research frame in the introduction section supported by education for sustainable development. Please check line 72-90 for the newly added part in the introduction to reflect upon education for sustainable development and see below the statements used in the revised manuscript:

“Based on the results of this study, school principals are expected to support the development of teachers using motivating language within the appropriate cultural context in order to contribute to education for sustainable development. Despite the fact that a significant body of literature and discussion surrounding education for sustainable development exists, the language and linguistic communication skills school principals have to sustain the motivation of teachers and enhance their self-efficacy has received scant attention. In our study, we partially address this gap in the literature by clearly highlighting the aspects of motivational language use by school principals to enhance teachers’ self-efficacy in the light of cultural context. School principals’ influence on teachers through motivational language use is substantial as the level of teachers’ self-efficacy increases, teachers’ burn out and attrition can be lessened and hence, this may result in more motivated, committed and sustained workforce of teachers as well as higher students’ academic achievement [17,18] and teaching effectiveness [19] that are all likely to contribute to education for sustainable development. While education for sustainable development enables people to understand the influences of their own actions by encouraging them to take responsible decision [20], enhancing teachers’ self-efficacy through motivational language use by school principals can serve this purpose as teachers with high self-efficacy hold the sense of being able to achieve a certain task in a specific situation. More specifically, teachers with high self-efficacy believe that they can teach even the most difficult and unmotivated students so that they can improve student achievement [21].”

Point 3: The methodology of the research is well-defined and methods used properly. Outcomes are clearly stated. Nevertheless, it is difficult to understand how the principals were included in the data. As far as I can understand, the data was collected from the teachers, not the principals. How is the motivation language of the principals investigated? From the teachers?

I find it really difficult to read this from the manuscript, since the data collection and the sample are narrowly explained. For instance, if the Motivational language of the principals were asked from the teachers (with the ML scale), why the questions were addressed to the teachers (e.g. the given example ‘I clarify what behaviours are expected at work’). Please, explain the data collection more thoroughly.

As it is, I think that the causal impact of the ML on the self-efficacy is difficult to prove. I think this needs to be clarified and discussed in the research also more generally.

Response 3:  We made further explanation about how the school principals were included in the data and how the motivating language of the principals is investigated in the study. We have extended and redesigned the method section. We provided further details about sample, data collection and how questionnaires were administered. We have added several new headings before “measures” section such as “Design”, “Sample”, “Data Analyses”. We put a specific heading to explain data analysis after the heading of measures in line with the reviewer’s suggestion. We have added the following statements to the end of 3.2 Sample section (line 358-360) to explain the context where the study was conducted in order to understand the role of principals. We have included the average age, standard deviation; teacher experience and type of school in Section 3.2 Sample (line 352-360). Additionally, more information on the socio-demographic variables of the sample were added and shown in Table 5 (line 361).

Thanks to Reviewer 2 for raising the question addressed to the teachers (e.g. ‘I clarify what behaviours are expected at work’). It was a typo error and we have corrected this statement as such: “Provides me with easily understandable instructions about my work.” (please kindly see line 380).

We agree with the Reviewer 2 about the causal link between ML and self-efficacy that is hard to prove. Based on Bandura’s (1997) conceptualization, we have added a part to Section 2.3. ‘Relationship between Motivating Language and Self-Efficacy’ where we discussed this link. We have also elaborated on this relationship throughout manuscript, particularly in the discussion section (see line 468-485, 495-509) Please kindly check line 174-184 for the newly added part to support this relationship and see below the statements used in the revised manuscript:

“Bandura [23] conceptualized self-efficacy beliefs through a four-dimensional model (mastery experiences, verbal persuasion, vicarious experiences, and physiological arousal). From the teacher’s perspectives, the verbal persuasion “has to do with verbal interactions that a teacher receives about his or her performance and prospects for success from important others in the teaching context, such as administrators, colleagues, parents, and members of the community at large” [39] (p.945). The research on self-efficacy [1, 2, 40] has pointed to the particular importance of language and verbal persuasion among the four sources of self-efficacy, stating that ML might play a role in developing self-efficacy, inasmuch as people tend to exert more effort when they are encouraged in real terms [41] (p.365). Another study found that verbal persuasion increased the self-efficacy of pre-service teachers by means of representative experience [42].”

Point 4:

  1. Finally, I encourage the authors to broaden the description of the convenience samplingof the study. The description is so scarce, that the nature and context of the data are not evident. The sample (teachers) are described before the analysis (section 4, results), but how about the schools they and the principals were working? For the generalizations it would be important to describe the schools in more detail (e.g. how many teacher worked in the public / private sectors, app. how large to schools were, how many of them were elementary, secondary or high schools).
  2. For the outcomes, it would be important to know if the participants worked at the same schools. Clusters might affect the outcomes. Can you estimate the school-level effects with the multilevel models?   

The information on the principals’ motivational language use should be explained more thoroughly and the sampling should be described in more detail.  

Response 4:

  1. We have extended and rearranged the method section overall where we have described the school types participants are working (please check line 358-360) and created a seperate table in the sample section (please check line 361) where we have included the school types. The school principals’ motivational language was explained in-depth with the supporting evidence derived from the current research as well as prior studies throughout the manuscript. The sampling was described in greater details in sampling subsection of methods (line 352).
  2. We would like to thank Reviewer 2 for such a relevant and insightful suggestion. The current study design would not allow us to look for a multilevel model. However, we have incorporated this point as a future research suggestion at the end of Section 6 given as follows (line 599-601):

“Lastly, we did not consider whether the participants worked at the same schools. Clusters might affect the outcomes. It would be beneficial to investigate the school-level effects through multilevel models”.

Reviewer 3 Report

This article is very well written, coherent, understandable and easy to follow.

The introduction is very clear to the reader, both from the importance of communication and the motivation and objectives of the study. The particular use of language to motivate is also well argued. In my opinion, some aspects such as self-efficacy are difficult to measure, as they are rather subjective and changeable values, but in this study the process followed is correct and well argued.

The Literature review is also well described in its entirety. However, in case there would be another version of this article, I would recommend some improvements. For example, references from authors such as Searle, Austin or Habermas, who were pioneers in speech acts, verbal and non-verbal communication and the importance of going beyond words, to perceive gestures, body language, and other acts. I perfectly understand that authors use Sullivan to describe perlocutionary, locutionary and illocutionary acts, which is fine, however, the most famous scholar in these terms is still considered to be Austin. In the same line, the Habermas Theory of Communicative Action is still considered a classical contribution to the field, as he mentioned the power of understanding, in line with what the article refers to in the cultural context. Even so, the article contains relevant literature, well explained and with the most recent studies quoted.

There is another point I would like to raise here. The link between individualistic and communication is somewhat questionable within research, as other studies might show just the opposite, but it is well quoted here and connected to the text.

Regarding the hypothesis, I consider all of them are well constructed. The H3 actually show how more than a factor is involved to determine self-efficacy perceptions.

Methodology section is very well explained. I would only have one suggestion; I would say that the first paragraph of the Results section, better goes on the Methods section as it describes participant’s profiles.

From the results part, I would like to highlight the motivational aspect, as interesting. Will motivation be more relevant than language, in relation to self-efficacy?

Another aspect that I consider important to highlight has to do with the objective of implementing ML. Could this goal have anything to do with student learning? On the implication paragraph, on the lines connected with improving self-efficacy, I would say, why? Might this improvement be connected with pupils’ academic performance?

It would be interesting to see the impact of all this on children’s academic performance, although I already understand that this is not the aim of this study.

Research has already shown that teachers' expectations are good for academic achievement. It would be great if this point goes clearer, meaning that ML is not only to motivate the teachers but also to improve the student’s learning, through the motivation of the teachers, which in this case is also given by the motivation of the principal. This link would enrich an already very good paper.

Author Response

Point 1: The introduction is very clear to the reader, both from the importance of communication and the motivation and objectives of the study. The particular use of language to motivate is also well argued. In my opinion, some aspects such as self-efficacy are difficult to measure, as they are rather subjective and changeable values, but in this study the process followed is correct and well argued.

Regarding the hypothesis, I consider all of them are well constructed. The H3 actually show how more than a factor is involved to determine self-efficacy perceptions.

Response 1: We would like to thank Reviewer 3 for overall evaluation about our study.

Point 2: The Literature review is also well described in its entirety. However, in case there would be another version of this article, I would recommend some improvements. For example, references from authors such as Searle, Austin or Habermas, who were pioneers in speech acts, verbal and non-verbal communication and the importance of going beyond words, to perceive gestures, body language, and other acts. I perfectly understand that authors use Sullivan to describe perlocutionary, locutionary and illocutionary acts, which is fine, however, the most famous scholar in these terms is still considered to be Austin. In the same line, the Habermas Theory of Communicative Action is still considered a classical contribution to the field, as he mentioned the power of understanding, in line with what the article refers to in the cultural context. Even so, the article contains relevant literature, well explained and with the most recent studies quoted.

Response 2: Thanks to Reviewer 3 for his/her insightful suggestion to improve the paper. In line with this suggestion, we have incorporated Searle, Austin and Habermas who are the leading authors and pioneers in speech acts, verbal and non-verbal communication. Please kindly check line 219-221 for the newly added part to Section 2.4. Cultural Context and see below the statements used in the revised manuscript:

“Actually, the role of the context of a speech act considered as constructed has taken its roots in the earlier and pioneering studies in the philosophy of language [54,55,56]”.

Point 3: Methodology section is very well explained. I would only have one suggestion; I would say that the first paragraph of the Results section, better goes on the Methods section as it describes participant’s profiles.

Response 3:  The first paragraph of the results section has now been moved to the methods section where the profile of participants is described in greater details. We have extended and rearranged the method section by dividing into several headings (please check line 342-463 for the entire method section)

  1. Method

3.1. Design

3.2. Sample

3.3. Data Analyses

3.3.1. Common Method Variance

3.3.2. Measures

3.3.2.1. Validation of the Measures

3.3.3. Results of Hypothesis Tests

We provided further details about sample, data collection and how questionnaires were administered. We have added several new headings before “measures” section such as “Design”, “Sample”, “Data Analyses”. We put a specific heading to explain data analysis after the heading of measures in line with the reviewer’s suggestion. We have added the following statements to the end of 3.2 Sample section (line 358-360) to explain the context where the study was conducted in order to understand the role of principals. We have included the average age, standard deviation; teacher experience and type of school in Section 3.2 Sample (line 352-360). Additionally, more information on the socio-demographic variables of the sample were added and shown in Table 5 (line 361).

Point 4: Another aspect that I consider important to highlight has to do with the objective of implementing ML. Could this goal have anything to do with student learning? On the implication paragraph, on the lines connected with improving self-efficacy, I would say, why? Might this improvement be connected with pupils’ academic performance?

It would be interesting to see the impact of all this on children’s academic performance, although I already understand that this is not the aim of this study.

Research has already shown that teachers' expectations are good for academic achievement. It would be great if this point goes clearer, meaning that ML is not only to motivate the teachers but also to improve the student’s learning, through the motivation of the teachers, which in this case is also given by the motivation of the principal. This link would enrich an already very good paper.

Response 4: Thanks to the reviewer 3 for raising an excellent point that is worth considering. Yes, motivating language is linked to student learning. In order to elaborate more on the issue whether improving teachers’ self-efficacy with school principals’ motivational language is connected to pupils’ academic performance, we revisited the literature thoroughly and incorporated several important findings from the literature to support this claim.

Please kindly check line 163-169 to support this relationship and see below the statements used in the revised manuscript:

“There is a large body of evidence supported by multivariate meta-analyses on teachers’ self-efficacy. Accordingly, classroom management self-efficacy of teachers was significantly and negatively associated with burnout [21; 37] and positively associated with students’ academic achievement (17). Teachers’ self-efficacy was not only found to enhance students’ academic achievement but also overall teaching effectiveness (19). In predicting commitment to the teaching profession of preservice and inservice teachers, their self-efficacy beliefs can be utilized [38]”.

Please kindly check line 479-485 to support this relationship and see below the statements used in the revised manuscript:

“The literature [39] regarding teacher self-efficacy found a strong relationship with ‘many meaningful, sustainable educational outcomes such as teachers’ persistence, enthusiasm, commitment, and instructional behaviour, as well as student outcomes such as achievement, motivation, and self-efficacy beliefs’ (p.783). On the other hand, the substantial evidence showing the power of motivational variables on teaching effectiveness is even greater than the effect of personality traits [79]. Thus, it is essential to keep searching for the factors that may increase teachers’ perception of their self-efficacy beliefs”.

Please kindly check line 516-522 to support this relationship and see below the statements used in the revised manuscript:

“The literature [39] regarding teacher self-efficacy found a strong relationship with ‘many meaningful, sustainable educational outcomes such as teachers’ persistence, enthusiasm, commitment, and instructional behaviour, as well as student outcomes such as achievement, motivation, and self-efficacy beliefs’ (p.783). On the other hand, the substantial evidence showing the power of motivational variables on teaching effectiveness is even greater than the effect of personality traits [79]. Thus, it is essential to keep searching for the factors that may increase teachers’ perception of their self-efficacy beliefs”.

Point 5: There is another point I would like to raise here. The link between individualistic and communication is somewhat questionable within research, as other studies might show just the opposite, but it is well quoted here and connected to the text.

From the results part, I would like to highlight the motivational aspect, as interesting. Will motivation be more relevant than language, in relation to self-efficacy?

Response 5: Thanks to Reviewer 3 for highlighting a different perspective. Please kindly check line 523–531 for the newly added paragraph to address the reviewer’s suggestion and see below the statements used in the revised manuscript:

“Other antecedents of general self-efficacy of employees exist in the literature. For example, the causal relationship has been founded regarding the role of self-efficacy on motivating individuals. The other way of such relationship has been rarely subject to scholarly investigation. By referring to Lunenburg [80] argues that goal-setting theory and self-efficacy theory are complementing each other. Setting difficult goals for employees by their managers could lead them to have a higher level of self-efficacy. But the issue whether each employee react the same to difficult tasks and motivated is still questionable. Here, then, ML could be resourceful for managers when they confront with this type of case. If they realize that the employee is about to give up, they could use ML to encourage them to not to give up and continue”.

Round 2

Reviewer 1 Report

This manuscript is an excellent contribution to the literature on the relationship between teachers' perception of school principals’ motivating language and self-efficacy, using a  Motivating language theory (MLT) as a theoretical framework. I was surprised by the quality of the authors’ revision effort. Frankly, I thought the reviewers were asking too much of the authors to handle/revise, but the author team carried out an excellent revision. Specifically, the authors have strengthened the introduction and discussion section. My personal recommendation is to accept the manuscript without revision.

Reviewer 2 Report

This manuscript has been revised and the major revisions made. The core focus on education for sustainable development is yet somewhat general, but the concepts are used in coherent manner. Especially the methodology of the research is now thoroughly explained and introduced. The novelty and limits of the research are now more expressed. The outcomes are interesting and add the scholarly knowledge on Motivation Language and Self-Efficacy. In my opinion, the article can now be published a peer-reviewed scientific article.